# GRAF: Generative Radiance Fields for 3D-Aware Image Synthesis

**Katja Schwarz**[*]    **Yiyi Liao**[*]    **Michael Niemeyer**    **Andreas Geiger**
Autonomous Vision Group
MPI for Intelligent Systems and University of Tübingen
`{firstname.lastname}@tue.mpg.de`

## Abstract

While 2D generative adversarial networks have enabled high-resolution image synthesis, they largely lack an understanding of the 3D world and the image formation process. Thus, they do not provide precise control over camera viewpoint or object pose. To address this problem, several recent approaches leverage intermediate voxel-based representations in combination with differentiable rendering. However, existing methods either produce low image resolution or fall short in disentangling camera and scene properties, e.g., the object identity may vary with the viewpoint. In this paper, we propose a generative model for radiance fields which have recently proven successful for novel view synthesis of a single scene. In contrast to voxel-based representations, radiance fields are not confined to a coarse discretization of the 3D space, yet allow for disentangling camera and scene properties while degrading gracefully in the presence of reconstruction ambiguity. By introducing a multi-scale patch-based discriminator, we demonstrate synthesis of high-resolution images while training our model from unposed 2D images alone. We systematically analyze our approach on several challenging synthetic and real-world datasets. Our experiments reveal that radiance fields are a powerful representation for generative image synthesis, leading to 3D consistent models that render with high fidelity.

## 1   Introduction

Convolutional generative adversarial networks have demonstrated impressive results in synthesizing high-resolution images [6, 26, 34, 53] from unstructured image collections. However, despite this success, state-of-the-art models struggle to properly disentangle the underlying generative factors including 3D shape and viewpoint. This is in stark contrast to humans who have the remarkable ability to reason about the 3D structure of the world and imagine objects from novel viewpoints.

As reasoning in 3D is fundamental for applications in robotics, virtual reality or data augmentation, several recent works consider the task of *3D-aware image synthesis* [19, 39, 76], aiming at photo-realistic image generation with explicit control over the camera pose. In contrast to 2D generative adversarial networks, approaches for 3D-aware image synthesis learn a 3D scene representation which is explicitly mapped to an image using differentiable rendering techniques, hence providing control over both, scene content and viewpoint. Since 3D supervision or posed images are often hard to obtain in practice, recent works try to solve this task using 2D supervision only [19, 39]. Towards this goal, existing approaches generate discretized 3D representations, i.e., a voxel-grid representing either the full 3D object [19] or intermediate 3D features [39] as illustrated in Fig. 1. While modeling the 3D object in color space allows for exploiting differentiable rendering, the cubic memory growth of voxel-based representations limits [19] to low resolution and results in visible artifacts. Intermediate 3D features [39] are more compact and scale better with image resolution.

---

[*] Joint first authors with equal contribution.

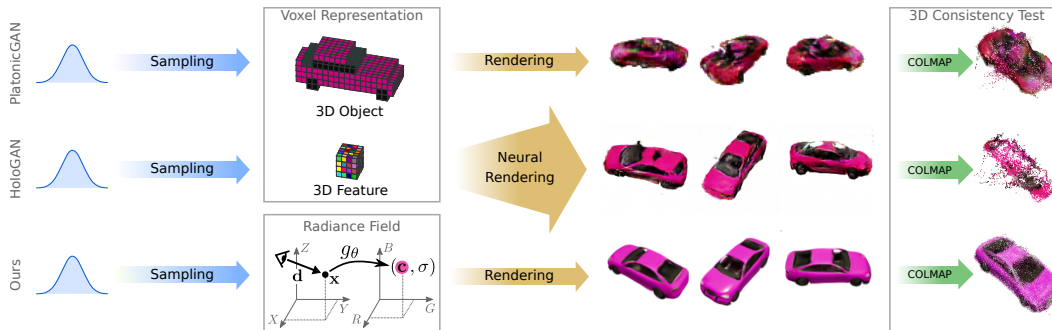

Figure 1: **Motivation.** Voxel-based approaches for 3D-aware image synthesis either generate a voxelized 3D model (e.g., PlatonicGAN [19], top) or learn an abstract 3D feature representation (e.g., HoloGAN [39], middle). This leads to discretization artifacts or degrades view-consistency of the generated images due to the learned neural projection function. In this paper, we propose a generative model for neural radiance fields (bottom) which represent the scene as a continuous function $g_\theta$ that maps a location $\mathbf{x}$ and viewing direction $\mathbf{d}$ to a color value $\mathbf{c}$ and a volume density $\sigma$. Our model allows for generating 3D consistent images at high spatial resolution. We visualize 3D consistency by running a multi-view stereo algorithm (COLMAP [60]) on several outputs of each method (right). Note that all models have been trained using 2D supervision only (i.e., from unposed RGB images).

However, this requires to *learn* a 3D-to-2D mapping for decoding the abstract features to RGB values, resulting in entangled representations which are not consistent across views at high resolutions.

In this paper, we demonstrate that the dilemma between coarse outputs and entangled latents can be resolved using conditional radiance fields, a conditional variant of a recently proposed continuous representation for novel view synthesis [36]. More specifically, we make the following contributions: i) We propose GRAF, a generative model for radiance fields for high-resolution 3D-aware image synthesis from unposed images. In addition to viewpoint manipulations, our approach allows to modify shape and appearance of the generated objects. ii) We introduce a patch-based discriminator that samples the image at multiple scales and which is key to learn high-resolution generative radiance fields efficiently. iii) We systematically evaluate our approach on synthetic and real datasets. Our approach compares favorably to state-of-the-art methods in terms of visual fidelity and 3D consistency while generalizing to high spatial resolutions. We release our code and datasets at https://github.com/autonomousvision/graf.

## 2 Related Work

**Image Synthesis:** Generative Adversarial Networks (GANs) [16, 51] have significantly advanced the state-of-the-art in photorealistic image synthesis. In order to make the image synthesis process more controllable, several recent works have proposed to disentangle the underlying factors of variation [9, 25, 28, 41, 54, 74]. However, all of these methods ignore the fact that 2D images are obtained as projections of the 3D world. While some of the methods demonstrate that the disentangled factors capture 3D properties to some extend [9, 28, 41], modeling the image manifold using 2D convolutional networks remains a difficult task, in particular when seeking representations that faithfully disentangle viewpoint variations from object appearance and identity. Instead of directly modeling the 2D image manifold, we therefore follow a recent line of works [19, 30, 39] which aims at generating 3D representations and explicitly models the image formation process.

**3D-Aware Image Synthesis:** Learning-based novel view synthesis has been intensively investigated in the literature [14, 36, 56, 61–64, 69, 75]. These methods generate unseen views from *the same* object and typically require camera viewpoints as supervision. While some works [14, 56, 62–64, 69] generalize across *different* objects without requiring to train an individual network per object, they do not yield a full probabilistic generative model for drawing unconditional random samples. In contrast, we are interested in generating *novel objects* from multiple views by learning a 3D-aware generative model from unposed 2D images.

Several recent works exploit generative 3D models for 3D-aware image synthesis [4,30,39,45,68,76]. Many methods require 3D supervision [68,76] or assume 3D information as input [4,45]. E.g. Texture Fields [45] synthesize novel textures conditioned on a particular 3D shape. Consequently, they require a 3D shape as input and colored surface points as supervision. Instead, we learn a generative model for both shape and texture *from 2D images alone*. This is a difficult task which only few works have attempted so far: PLATONICGAN [19] learns a textured 3D voxel representation from 2D images using differentiable rendering techniques. However, such voxel-based representations are memory intensive, precluding image synthesis at high image resolutions. In this work, we avoid these memory limitations by using a continuous representation which allows for rendering images at arbitrary resolution. HoloGAN [39] and some related works [30,40] learn a low-dimensional 3D feature combined with a learnable 3D-to-2D projection. However, as evidenced by our experiments, learned projections can lead to entangled latents (e.g., object identity and viewpoint), particularly at high resolutions. While 3D consistency can be encouraged using additional constraints [44], we take advantage of differentiable volume rendering techniques which do not need to be learned and thus incorporate 3D consistency into the generative model *by design*.

**Implicit Representations:** Recently, implicit representations of 3D geometry have gained popularity in learning-based 3D reconstruction [10,15,35,42,45,48,59]. Key advantages over voxel [7,11,55,57,70,71] or mesh-based [17,29,47,67] methods are that they do not discretize space and are not restriced in topology. Recent hybrid continuous grid representations [8,22,50] extend implicit representations to complicated or large scale scenes but require 3D input and do not consider texture. Another line of works [43,62,72] propose to learn continuous shape and texture representations from posed multi-view images only, by making the rendering process differentiable. As these models are limited to single objects or scenes of small geometric complexity, Mildenhall et al. [36] propose to represent scenes as neural radiance fields which allow for multi-view consistent novel-view synthesis of more complex, real-world scenes from posed 2D images. They demonstrate compelling results on this task, however, their method requires many posed views, needs to be retrained for each scene, and cannot generate novel scenes. Inspired by this work, we exploit a conditional variant of this representation and show how a rich generative model can be learned from a collection of unposed 2D images as input.

## 3 Method

We consider the problem of 3D-aware image synthesis, i.e., the task of generating high-fidelity images while providing explicit control over camera rotation and translation. We argue for representing a scene by its radiance field as such a continuous representation scales well wrt. image resolution and memory consumption while allowing for a physically-based and parameter-free projective mapping. In the following, we first briefly review Neural Radiance Fields (NeRF) [36] which forms the basis for the proposed Generative Radiance Field (GRAF) model.

### 3.1 Neural Radiance Fields

**Neural Radiance Fields:** A radiance field is a continuous mapping from a 3D location and a 2D viewing direction to an RGB color value [23,33]. Mildenhall et al. [36] proposed to use neural networks for representing this mapping. More specifically, they first map a 3D location $\mathbf{x} \in \mathbb{R}^3$ and a viewing direction $\mathbf{d} \in \mathbb{S}^2$ to a higher-dimensional feature representation using a fixed positional encoding which is applied element-wise to all three components of $\mathbf{x}$ and $\mathbf{d}$:

$$\gamma(p) = \left( \sin(2^0 \pi p), \cos(2^0 \pi p), \sin(2^1 \pi p), \cos(2^1 \pi p), \sin(2^2 \pi p), \cos(2^2 \pi p), \dots \right) \quad (1)$$

Following recent implicit models [10,35,48], they then apply a multi-layer perceptron $f_\theta(\cdot)$ with parameters $\theta$ for mapping the resulting features to a color value $\mathbf{c} \in \mathbb{R}^3$ and a volume density $\sigma \in \mathbb{R}^+$:

$$f_\theta : \mathbb{R}^{L_\mathbf{x}} \times \mathbb{R}^{L_\mathbf{d}} \to \mathbb{R}^3 \times \mathbb{R}^+ \qquad (\gamma(\mathbf{x}), \gamma(\mathbf{d})) \mapsto (\mathbf{c}, \sigma) \quad (2)$$

As demonstrated in [36,52], the positional encoding $\gamma(\cdot)$ enables better fitting of high-frequency signals compared to directly using $\mathbf{x}$ and $\mathbf{d}$ as input to the multi-layer perceptron $f_\theta(\cdot)$. We confirm this with an ablation study in our supp. material. As the volume color $\mathbf{c}$ varies more smoothly with the viewing direction than with the 3D location, the viewing direction is typically encoded using fewer components, i.e., $L_\mathbf{d} < L_\mathbf{x}$.

**Volume Rendering:** For rendering a 2D image from the radiance field $f_\theta(\cdot)$, Mildenhall et al. [36] approximate the intractable volumetric projection integral using numerical integration. More formally, let $\{(\mathbf{c}_r^i, \sigma_r^i)\}_{i=1}^N$ denote the color and volume density values of $N$ random samples along a camera ray $r$. The rendering operator $\pi(\cdot)$ maps these values to an RGB color value $\mathbf{c}_r$:

$$\pi : (\mathbb{R}^3 \times \mathbb{R}^+)^N \mapsto \mathbb{R}^3 \qquad \{(\mathbf{c}_r^i, \sigma_r^i)\} \mapsto \mathbf{c}_r \tag{3}$$

The RGB color value $\mathbf{c}_r$ is obtained using alpha composition

$$\mathbf{c}_r = \sum_{i=1}^N T_r^i \alpha_r^i \mathbf{c}_r^i \qquad T_r^i = \prod_{j=1}^{i-1} \left(1 - \alpha_r^j\right) \qquad \alpha_r^i = 1 - \exp\left(-\sigma_r^i \delta_r^i\right) \tag{4}$$

where $T_r^i$ and $\alpha_r^i$ denote the transmittance and alpha value of sample point $i$ along ray $r$ and $\delta_r^i = \left\| \mathbf{x}_r^{i+1} - \mathbf{x}_r^i \right\|_2$ is the distance between neighboring sample points. Given a set of posed 2D images of a single static scene, Mildenhall et al. [36] optimize the parameters $\theta$ of the neural radiance field $f_\theta(\cdot)$ by minimizing a reconstruction loss (sum of squared differences) between the observations and the predictions. Given $\theta$, novel views can be synthesized by invoking $\pi(\cdot)$ for each pixel/ray.

## 3.2 Generative Radiance Fields

In this work, we are interested in radiance fields as a representation for 3D-aware image synthesis. In contrast to [36], we do not assume a large number of posed images of a single scene. Instead, we aim at learning a model for synthesizing novel scenes by training on unposed images. More specifically, we utilize an adversarial framework to train a generative model for radiance fields (GRAF).

Fig. 2 shows an overview over our model. The generator $G_\theta$ takes camera matrix $\mathbf{K}$, camera pose $\boldsymbol{\xi}$, 2D sampling pattern $\boldsymbol{\nu}$ and shape/appearance codes $\mathbf{z}_s \in \mathbb{R}^m / \mathbf{z}_a \in \mathbb{R}^n$ as input and predicts an image patch $\mathbf{P}'$. The discriminator $D_\phi$ compares the synthesized patch $\mathbf{P}'$ to a patch $\mathbf{P}$ extracted from a real image $\mathbf{I}$. At inference time, we predict one color value for every image pixel. However, at training time, this is too expensive. Therefore, we instead predict a fixed patch of size $K \times K$ pixels which is randomly scaled and rotated to provide gradients for the entire radiance field.

### 3.2.1 Generator

We sample the camera pose $\boldsymbol{\xi} = [\mathbf{R}|\mathbf{t}]$ from a pose distribution $p_\xi$. In our experiments, we use a uniform distribution on the upper hemisphere for the camera location with the camera facing towards the origin of the coordinate system. Depending on the dataset, we also vary the distance of the camera from the origin uniformly. We choose $\mathbf{K}$ such that the principle point is in the center of the image.

$\boldsymbol{\nu} = (\mathbf{u}, s)$ determines the center $\mathbf{u} = (u, v) \in \mathbb{R}^2$ and scale $s \in \mathbb{R}^+$ of the virtual $K \times K$ patch $\mathcal{P}(\mathbf{u}, s)$ which we aim to generate. This enables us to use a convolutional discriminator independent of the image resolution. We randomly draw the patch center $\mathbf{u} \sim \mathcal{U}(\Omega)$ from a uniform distribution over the image domain $\Omega$ and the patch scale $s$ from a uniform distribution $s \sim \mathcal{U}([1, S])$ where $S = \min(W, H)/K$ with $W$ and $H$ denoting the width and height of the target image. Moreover, we ensure that the entire patch is within the image domain $\Omega$. The shape and appearance variables $\mathbf{z}_s$ and $\mathbf{z}_a$ are drawn from shape and appearance distributions $\mathbf{z}_s \sim p_s$ and $\mathbf{z}_a \sim p_a$, respectively. In our experiments we use a standard Gaussian distribution for both $p_s$ and $p_a$.

**Ray Sampling:** The $K \times K$ patch $\mathcal{P}(\mathbf{u}, s)$ is determined by a set of 2D image coordinates

$$\mathcal{P}(\mathbf{u}, s) = \left\{ (sx + u, sy + v) \,\middle|\, x, y \in \left\{ -\frac{K}{2}, \ldots, \frac{K}{2} - 1 \right\} \right\} \tag{5}$$

which describe the location of every pixel of the patch in the image domain $\Omega$ as illustrated in Fig. 3. Note that these coordinates are real numbers, not discrete integers which allows us to continuously evaluate the radiance field. The corresponding 3D rays are uniquely determined by $\mathcal{P}(\mathbf{u}, s)$, the camera pose $\boldsymbol{\xi}$ and intrinsics $\mathbf{K}$. We denote the pixel/ray index by $r$, the normalized 3D rays by $\mathbf{d}_r$ and the number of rays by $R$ where $R = K^2$ during training and $R = WH$ during inference.

**3D Point Sampling:** For numerical integration of the radiance field, we sample $N$ points $\{\mathbf{x}_r^i\}$ along each ray $r$. We use the stratified sampling approach of [36], see supp. material for details.

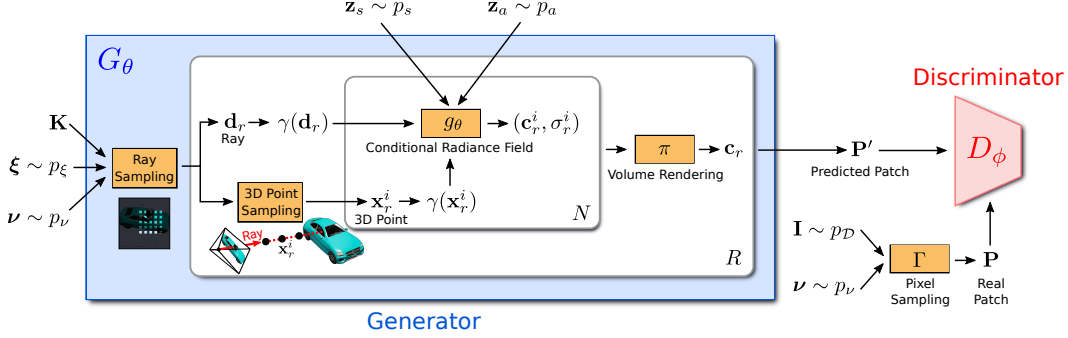

Figure 2: **Generative Radiance Fields.** The generator $G_\theta$ takes camera matrix $\mathbf{K}$, camera pose $\boldsymbol{\xi}$, 2D sampling pattern $\boldsymbol{\nu}$ and shape/appearance codes $\mathbf{z}_s \in \mathbb{R}^m / \mathbf{z}_a \in \mathbb{R}^n$ as input and predicts an image patch $\mathbf{P}'$. We use plate notation to illustrate $R$ rays and $N$ samples per ray. Note that the conditional radiance field $g_\theta$ is the only component with trainable parameters. The discriminator $D_\phi$ compares the synthesized patch $\mathbf{P}'$ to a real patch $\mathbf{P}$ extracted from a real image $\mathbf{I}$. At training time, we use sparse 2D sampling patterns of size $K \times K$ pixels for computational and memory efficiency. At inference time, we predict one color value for every pixel in the target image.

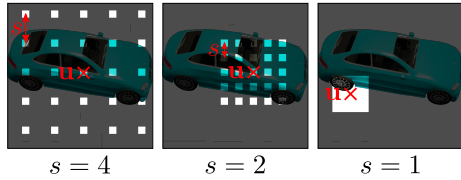

Figure 3: **Ray Sampling.** Given camera pose $\boldsymbol{\xi}$, we sample rays according to $\boldsymbol{\nu} = (\mathbf{u}, s)$ which determines the continuous 2D translation $\mathbf{u} \in \mathbb{R}^2$ and scale $s \in \mathbb{R}^+$ of a $K \times K$ patch. This enables us to use a convolutional discriminator independent of the image resolution.

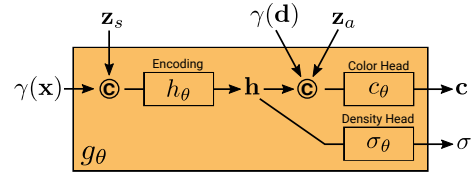

Figure 4: **Conditional Radiance Field.** While the volume density $\sigma$ depends solely on the 3D point $\mathbf{x}$ and the shape code $\mathbf{z}_s$, the predicted color value $\mathbf{c}$ additionally depends on the viewing direction $\mathbf{d}$ and the appearance code $\mathbf{z}_a$, modeling view-dependent appearance, e.g., specularities.

**Conditional Radiance Field:** The radiance field is represented by a deep fully-connected neural network with parameters $\theta$ that maps the positional encoding (cf. Eq. (1)) of 3D location $\mathbf{x} \in \mathbb{R}^3$ and viewing direction $\mathbf{d} \in \mathbb{S}^2$ to an RGB color value $\mathbf{c}$ and a volume density $\sigma$:

$$g_\theta : \mathbb{R}^{L_{\mathbf{x}}} \times \mathbb{R}^{L_{\mathbf{d}}} \times \mathbb{R}^{M_s} \times \mathbb{R}^{M_a} \to \mathbb{R}^3 \times \mathbb{R}^+ \qquad (\gamma(\mathbf{x}), \gamma(\mathbf{d}), \mathbf{z}_s, \mathbf{z}_a) \mapsto (\mathbf{c}, \sigma) \qquad (6)$$

Note that in contrast to (2), $g_\theta$ is conditioned on two additional latent codes: a shape code $\mathbf{z}_s \in \mathbb{R}^{M_s}$ which determines the shape of the object and an appearance code $\mathbf{z}_a \in \mathbb{R}^{M_a}$ which determines its appearance. We thus call $g_\theta$ a *conditional radiance field*.

The network architecture of our conditional radiance field $g_\theta$ is illustrated in Fig. 4. We first compute a shape encoding $\mathbf{h}$ from the positional encoding of $\mathbf{x}$ and the shape code $\mathbf{z}_s$. A density head $\sigma_\theta$ transforms this encoding to the volume density $\sigma$. For predicting the color $\mathbf{c}$ at 3D location $\mathbf{x}$, we concatenate $\mathbf{h}$ with the positional encoding of $\mathbf{d}$ and the appearance code $\mathbf{z}_a$ and pass the resulting vector to a color head $c_\theta$. We compute $\sigma$ independently of the viewpoint $\mathbf{d}$ and the appearance code $\mathbf{z}_a$ to encourage multi-view consistency while disentangling shape from appearance. This encourages the network to use the latent codes $\mathbf{z}_s$ and $\mathbf{z}_a$ to model shape and appearance, respectively, and allows for manipulating them separately during inference. More formally, we have:

$$h_\theta : \mathbb{R}^{L_{\mathbf{x}}} \times \mathbb{R}^{M_s} \to \mathbb{R}^H \qquad\qquad (\gamma(\mathbf{x}), \mathbf{z}_s) \mapsto \mathbf{h} \qquad (7)$$

$$c_\theta : \mathbb{R}^H \times \mathbb{R}^{L_{\mathbf{d}}} \times \mathbb{R}^{M_a} \to \mathbb{R}^3 \qquad (\mathbf{h}(\mathbf{x}, \mathbf{z}_s), \gamma(\mathbf{d}), \mathbf{z}_a) \mapsto \mathbf{c} \qquad (8)$$

$$\sigma_\theta : \mathbb{R}^H \to \mathbb{R}^+ \qquad\qquad \mathbf{h}(\mathbf{x}, \mathbf{z}_s) \mapsto \sigma \qquad (9)$$

All mappings ($h_\theta$, $c_\theta$ and $\sigma_\theta$) are implemented using fully connected networks with ReLU activations. To avoid notation clutter, we use the same symbol $\theta$ to denote the parameters of each network.

**Volume Rendering:** Given the color and volume density $\{(\mathbf{c}_r^i, \sigma_r^i)\}$ of all points along ray $r$, we obtain the color $\mathbf{c}_r \in \mathbb{R}^3$ of the pixel corresponding to ray $r$ using the volume rendering operator in Eq. (3). Combining the results of all $R$ rays, we denote the predicted patch as $\mathbf{P}'$ as shown in Fig. 2.

### 3.2.2 Discriminator

The discriminator $D_\phi$ is implemented as a convolutional neural network (see supp. material for details) which compares the predicted patch $\mathbf{P}'$ to a patch $\mathbf{P}$ extracted from a real image $\mathbf{I}$ drawn from the data distribution $p_\mathcal{D}$. For extracting a $K \times K$ patch from real image $\mathbf{I}$, we first draw $\boldsymbol{\nu} = (\mathbf{u}, s)$ from the same distribution $p_\nu$ which we use for drawing the generator patch above. We then sample the real patch $\mathbf{P}$ by querying $\mathbf{I}$ at the 2D image coordinates $\mathcal{P}(\mathbf{u}, s)$ using bilinear interpolation. In the following, we use $\Gamma(\mathbf{I}, \boldsymbol{\nu})$ to denote this bilinear sampling operation. Note that our discriminator is similar to PatchGAN [21], except that we allow for continuous displacements $\mathbf{u}$ and scales $s$ while PatchGAN uses $s = 1$. It is further important to note that we do not downsample the real image $\mathbf{I}$ based on $s$, but instead query $\mathbf{I}$ at sparse locations to retain high-frequency details, see Fig. 3.

Experimentally, we found that a single discriminator with shared weights is sufficient for all patches, even though these are sampled at random locations with different scales. Note that the scale determines the *receptive field* of the patch. To facilitate training, we thus start with patches of larger receptive fields to capture the global context. We then progressively sample patches with smaller receptive fields to refine local details.

### 3.2.3 Training and Inference

Let $\mathbf{I}$ denote an image from the data distribution $p_\mathcal{D}$ and let $p_\nu$ denote the distribution over random patches (see Section 3.2.1). We train our model using a non-saturating GAN loss with R1-regularization [34]

$$
\begin{aligned}
\mathcal{L}(\theta, \phi) = &\ \mathbb{E}_{\mathbf{z}_s \sim p_s, \mathbf{z}_a \sim p_a, \boldsymbol{\xi} \sim p_\xi, \boldsymbol{\nu} \sim p_\nu} \left[ f(D_\phi(G_\theta(\mathbf{z}_s, \mathbf{z}_a, \boldsymbol{\xi}, \boldsymbol{\nu}))) \right] \\
& + \mathbb{E}_{\mathbf{I} \sim p_\mathcal{D}, \boldsymbol{\nu} \sim p_\nu} \left[ f(-D_\phi(\Gamma(\mathbf{I}, \boldsymbol{\nu}))) + \lambda \| \nabla D_\phi(\Gamma(\mathbf{I}, \boldsymbol{\nu})) \|^2 \right]
\end{aligned}
\tag{10}
$$

where $f(t) = -\log(1 + \exp(-t))$ and $\lambda$ controls the strength of the regularizers. We use spectral normalization [37] and instance normalization [65] in our discriminator and train our approach using RMSprop [27] with a batch size of $8$ and a learning rate of $0.0005$ and $0.0001$ for generator and discriminator, respectively. At inference, we randomly sample $\mathbf{z}_s$, $\mathbf{z}_a$ and $\boldsymbol{\xi}$, and predict a color value for all pixels in the image. Details on the network architectures can be found in the supp. material.

## 4 Experiments

**Datasets:** We consider two synthetic and three real-world datasets in our experiments. To analyze our approach in a controlled setting we render 150k *Chairs* from Photoshapes [49] following the rendering protocol of [46]. We further use the Carla Driving simulator [12] to create 10k images of 18 car models with randomly sampled colors and realistic texture and reflectance properties (*Cars*). We also validate our approach on three real-world datasets. We use the *Faces* dataset which comprises celebA [31] and celebA-HQ [24] for image synthesis up to resolution $128^2$ and $512^2$ pixels, respectively. In addition, we consider the *Cats* dataset [73] and the Caltech-UCSD *Birds*-200-2011 [66] dataset. For the latter, we use the available instance masks to composite the birds onto a white background.

**Baselines:** We compare our approach to two state-of-the-art models for 3D-aware image synthesis using the authors' implementations[1][2]: PLATONICGAN [19] generates a voxel-grid of the 3D object which is projected to the image plane using differentiable volumetric rendering. HoloGAN [39] instead generates an abstract voxelized feature representation and learns the mapping from 3D to 2D using a combination of 3D and 2D convolutions. To analyze the consequences of a learned projection we further consider a modified version of HoloGAN (HoloGAN w/o 3D Conv) in which we reduce the capacity of the learned mapping by removing the 3D convolutional layers. For reference, we also compare our results to a state-of-the-art 2D GAN model [34] with a ResNet [18] architecture.

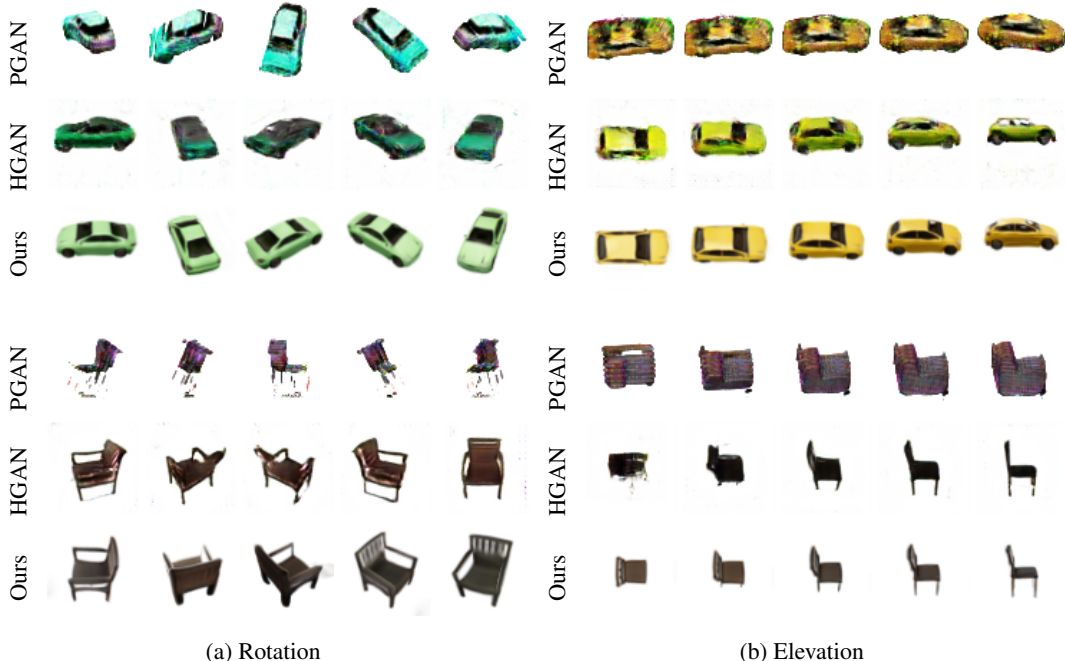

(a) Rotation                                                          (b) Elevation

Figure 5: **Camera Pose Interpolations** for Cars and Chairs at image resolution $64^2$ pixels for PLATONICGAN [19] (PGAN), HoloGAN [39] (HGAN) and our approach (Ours).

|  | Chairs | Birds | Cars | Cats | Faces |
|---|---|---|---|---|---|
| 2D GAN [34] | 59 | 24 | 66 | 18 | 15 |
| PLATONICGAN [19] | 199 | 179 | 169 | 318 | 321 |
| HoloGAN [39] | 59 | 78 | 134 | 27 | 25 |
| Ours | 34 | 47 | 30 | 26 | 25 |

Table 1: **FID** at image resolution $64^2$ pixels.

|  | Cars | | | Faces | | |
|---|---|---|---|---|---|---|
|  | 128 | 256 | 512 | 128 | 256 | 512 |
| HoloGAN [39] | 211 | 230 | – | 39 | 61 | – |
| w/o 3D Conv | 180 | 189 | 251 | 31 | 33 | 51 |
| Ours | 41 | 71 | 84 | 35 | 49 | 49 |
| upsampled | – | 91 | 128 | – | 63 | 77 |
| sampled | – | 74 | 104 | – | 50 | 56 |

Table 2: **FID** at image resolution $128^2$-$512^2$.

**Evaluation Metrics:** We quantify image fidelity using the Frechet Inception Distance (*FID*) [20] and additionally report the Kernel Inception Distance (*KID*) [5] in the supp. material. To assess 3D consistency we perform 3D reconstruction for images of size $256^2$ pixels using COLMAP [60]. We adopt Minimum Matching Distance (MMD) [1] to measure the chamfer distance (CD) between 100 reconstructed shapes and their closest shapes in the ground truth for quantitative comparison and show qualitative results for the reconstructions.

We now study several key questions relevant to the proposed model. We first compare our model to several baselines in terms of their ability to generate high-fidelity and high-resolution outputs. We then analyze the implications of learned projections and the importance of our multi-scale discriminator.

**How do Generative Radiance Fields compare to voxel-based approaches?** We first compare our model against the baselines using an image resolution of $64^2$ pixels. As shown in Fig. 5, all methods are able to disentangle object identity and camera viewpoint. However, PLATONICGAN has difficulties in representing thin structures and both PLATONICGAN and HoloGAN lead to visible artifacts in comparison to the proposed model. This is also reflected by larger FID scores in Table 1. On Faces and Cats, HoloGAN achieves FID scores similar to our approach as both datasets exhibit only little variation in the azimuth angle of the camera while the other datasets cover larger viewpoint variations. This suggests that it is harder for HoloGAN to accurately capture the appearance of objects from different viewpoints due to its low-dimensional 3D feature representation and the learnable

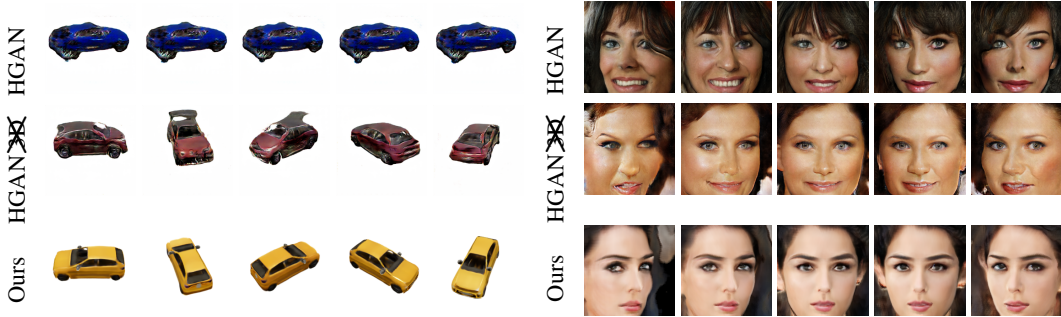

Figure 6: **Viewpoint Interpolations** on Faces and Cars at image resolution $256^2$ pixels for HoloGAN [39] (HGAN), HoloGAN w/o 3D Conv (HGAN⚡) and our approach (Ours).

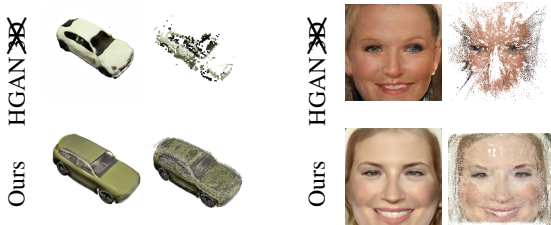

Figure 7: **3D Reconstruction** from synthesized images at resolution $256^2$. Each pair shows one of the generated images and the 3D reconstruction from COLMAP [60].

| Method | MMD-CD |
|---|---|
| Ours | **0.044** |
| HGAN | 0.109 |
| HGAN⚡ | 0.092 |

Table 3: **Reconstruction Accuracy** on Cars for 100 COLMAP reconstructions compared to their closest shapes in the ground truth in terms of MMD [1] measuring chamfer distance (CD).

projection. In contrast, our continuous representation does not require a learned projection and renders high-fidelity images from arbitrary views.

**Do 3D-aware generative models scale to high-resolution outputs?** Due to the voxel-based representation, PLATONICGAN becomes very memory intensive when scaled to higher resolutions. Thus, we limit our experiments to HoloGAN and HoloGAN w/o 3D Conv for this analysis. Additionally, we provide results for our model trained at $128^2$ pixels, but sampled at higher resolution during inference (*Ours sampled*). The results in Table 2 show that this significantly improves over naïve bilinear upsampling (*Ours upsampled*) which indicates that our learned representation generalizes well to higher resolutions. As expected, our method achieves the smallest FID value when trained at full resolution. While our approach outperforms HoloGAN and HoloGAN w/o 3D Conv significantly on the Car dataset, HoloGAN w/o 3D Conv achieves results onpar with us on Faces where viewpoint variations are smaller. Interestingly, we found that HoloGAN w/o 3D Conv achieves lower FID values than the full HoloGAN model originally proposed in [39] despite reduced model capacity. This is due to training instabilities which we observe when training HoloGAN at high resolutions. We even observe mode collapse at a resolution of $512^2$ for which we are hence not able to report results.

**Should learned projections be avoided?** As illustrated in Fig. 6, HoloGAN (top) fails in disentangling viewpoint from appearance at high resolution, varying different style aspects like facial expression or even completely ignoring the pose input. We identify the learnable projection as the underlying cause for this behavior. In particular, we find that removing the 3D convolutional layers enables HoloGAN to adhere to the input pose more closely, see Fig. 6 (middle). However, images from HoloGAN w/o 3D Conv are still not entirely multi-view consistent. To better investigate this observation we generate multiple images of the same instance at random viewpoints for both HoloGAN w/o 3D Conv and our approach, and perform dense 3D reconstruction using COLMAP [60]. As reconstruction depends on the consistency across views, reconstruction accuracy can be considered as a proxy for the multi-view consistency of the generated images. As evident from Table 3 and Fig. 7, multi-view stereo works significantly better when using images from our method as input. In contrast, fewer correspondences can be established for HoloGAN w/o 3D Conv which uses learned 2D layers for upsampling. For HoloGAN with 3D convolutions, performance degrades even further as shown in the supp. material. We thus conjecture that learned projections should generally be avoided.

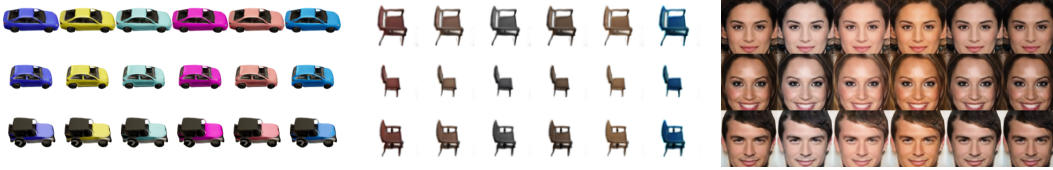

Figure 8: **Disentangling Shape / Appearance.** Results from our model on Cars, Chairs and Faces.

|  | Cars | CelebA |
|---|---|---|
| Full (64), dim/2 | 29 | 77 |
| Patch (64), dim/2 | 32 | 28 |
| Patch (128) | 41 | 35 |
| Patch (128), $s = 1$ | 277 | 153 |

Table 4: **Patch Sampling Strategies.** Ablation study comparing patch sampling strategies in terms of FID score.

| $f/f_{data}$ | FID | $f/f_{data}$ | FID |
|---|---|---|---|
| 0.3 | 84 | 1.1 | 48 |
| 0.5 | 61 | 1.2 | 57 |
| 0.7 | 40 | 1.3 | 47 |
| 0.8 | 36 | 1.5 | 52 |
| 0.9 | 38 | 1.8 | 51 |
| 1.0 | 41 | orthographic | 53 |

Table 5: **Focal Length.** Ablation on the choice of the focal length of the camera in terms of FID.

**Are Generative Radiance Fields able to disentangle shape from appearance?** Fig. 8 shows that in addition to disentangling camera and scene properties, our approach learns to disentangle shape and appearance which can be controlled during inference via $\mathbf{z}_s$ and $\mathbf{z}_a$. For Cars and Chairs the appearance code controls the color of the object while for Faces it encodes skin and hair color.

**How important is the multi-scale patch discriminator?** To investigate whether we sacrifice image quality by using the proposed multi-scale patch discriminator, we compare our multi-scale discriminator (*Patch*) to a discriminator that receives the entire image as input (*Full*). As this is very memory intensive, we only consider images of resolution $64^2$ and use half the hidden dimensions for $h_\theta$ and $c_\theta$ (dim/2). Table 4 shows that our patch discriminator achieves similar performance to the full image discriminator on Cars and performs even better on CelebA. A possible explanation for this phenomenon is that random patch sampling acts as a data augmentation strategy which helps to stabilize GAN training. In contrast, when using only local patches ($s = 1$), we observe that our generator is not able to learn the correct shape, resulting in a high FID value in Table 4. We conclude that sampling patches at random scales is crucial for robust performance.

**How important are the camera intrinsics?** We ablate the sensitivity of our model to the chosen focal length on Cars under a fixed radius of 10 in Table 5. Our model performs very similar for changes within $0.7 f_{data}$ to $1.0 f_{data}$ where $f_{data}$ is the focal length we use to render the training images. Even for larger focal lengths up to $1.8 f_{data}$ and with an orthographic projection we observe good performance. Only for very small focal lengths the generated images show distortions at the image borders resulting in higher FID values.

## 5   Conclusion

We have introduced Generative Radiance Fields (GRAF) for high-resolution 3D-aware image synthesis. We showed that our framework is able to generate high resolution images with better multi-view consistency compared to voxel-based approaches. However, our results are limited to simple scenes with single objects. We believe that incorporating inductive biases, e.g., depth maps or symmetry, will allow for extending our model to even more challenging real-world scenarios in the future.

## Broader Impact

3D-aware image synthesis is a relatively novel research area [19, 30, 39, 40, 44] and does not yet scale to generating complex real-world scenes, preventing immediate applications for society. However, our work takes an important step towards this goal as it enables high-fidelty reconstruction at resolutions beyond $64^2$ pixels while requiring no 3D supervision as input. In the long run, we hope that our resesarch will facilitate the use of 3D-aware generative models in applications such as virtual reality, data augmentation or robotics. For example, intelligent systems such as autonomous vehicles require large amounts of data for training and validation which will be impossible to collect using static offline datasets. We believe that building generative, photo-realistic and large-scale 3D models of our world will ultimately allow for cost-efficient data collection and simulation. While many use-cases are possible, we believe that these types of models can be particularly beneficial to close the existing domain gap between real-world and synthetic data. However, working with generative models also requires care. While generating photorealistic 3D-scenarios is very intriguing it also bears the risk of manipulation and the creation of misleading content. In particular, models that can create 3D-consistent fake images might increase credibility of fake contents and might potentially fool systems that rely on multi-view consistency, e.g., modern face recognition systems. Therefore, we believe that it is of equal importance for the community to develop methods which are able to clearly distinguish between synthetic and real-world content. We see encouraging progress in this area, e.g., [2, 3, 13, 32, 38, 58].

## Acknowledgments

We acknowledge the financial support by the BMWi in the project KI Delta Learning (project number 19A19013O) and the support from the BMBF through the Tuebingen AI Center (FKZ: 01IS18039A). We thank the International Max Planck Research School for Intelligent Systems (IMPRS-IS) for supporting Katja Schwarz and Michael Niemeyer. This work was supported by an NVIDIA research gift.

## Footnotes

[1] PLATONICGAN: https://github.com/henzler/platonicgan

[2] HoloGAN: https://github.com/thunguyenphuoc/HoloGAN

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
