[Supplementary Material · supplementary.pdf]

# Supplementary Material for
# GRAF: Generative Radiance Fields for 3D-Aware Image Synthesis

**Katja Schwarz**[*]   **Yiyi Liao**[*]   **Michael Niemeyer**   **Andreas Geiger**
Autonomous Vision Group
MPI for Intelligent Systems and University of Tübingen
Tübingen, 72076
{firstname.lastname}@tue.mpg.de

## Abstract

In this **supplementary document**, we first provide details on the network architectures of our approach and the baselines in Section 1. Section 2 describes the chosen hyperparameters for each dataset. In Section 3, we show additional results and failure cases and discuss limitations of our approach. The **supplementary video** shows synthesized animations in which we control the camera viewpoint and interpolate between latent codes. We use the same mathematical notation as in the paper.

## 1   Implementation

### 1.1   Generative Radiance Fields

The network architectures for our generator and our discriminator are specified in Table 1 and Table 2, respectively. Following [9], for $D_\phi$ instance normalization is applied to the features and spectral normalization is applied to the weights. For $g_\theta$ we compute the final weights as exponential moving average [15] with decay 0.999. In all experiments we set $m = n = 128$ and $L_\mathbf{x} = 3 \cdot 2 \cdot 10$, $L_\mathbf{d} = 3 \cdot 2 \cdot 4$ (i.e. we embed each of the three coordinates of $\mathbf{x}$ to dimension 10 and each of the three coordinates of $\mathbf{d}$ to dimension 4. The factor 2 comes from the two periodic functions used in the embedding, namely sinus and cosinus). For ray and point sampling we use $K = 32$ and $N = 64$, respectively. We exponentially decrease the minimal scale (that determines the minimal receptive field of the patches) as $s = \max\left(1, S\exp(-0.0025i)\right)$ where $i$ denotes the training iteration. We choose the camera matrix $\mathbf{K}$ and the camera's distance to the origin empirically such that objects approximately cover the entire image. As shown by our ablation study in Table 5 of the paper our model is relatively robust to different choices of $\mathbf{K}$.

**3D Point Sampling:**   To approximate the intractable volumetric projection integral Mildenhall et al. [8] propose a stratified sampling approach that allows to query the network at continuous intervals instead of a discretized grid. More specifically, they partition each ray $r$ into $N$ evenly-spaced bins and sample uniformly at random within each bin. Let $\mathbf{r}_r(\tau) = \mathbf{t} + \tau\mathbf{d}_r$ denote the straight line describing ray $r$ and let $\tau_n$, $\tau_f$ define near and far plane of the camera, then:

$$\mathbf{x}_r^i = \mathbf{r}_r(\tau_r^i) \qquad \tau_r^i \sim \mathcal{U}\left[\tau_n + \frac{i-1}{N}(\tau_f - \tau_n), \tau_n + \frac{i}{N}(\tau_f - \tau_n)\right] \tag{1}$$

---

[*] Joint first authors with equal contribution.

|            | Layer Type | Input Dimension | Output Dimension | Activation | Repetitions |
|------------|------------|-----------------|------------------|------------|-------------|
| $h_\theta$ | Linear     | $L_\mathbf{x} + m = 189$ | 256  | Relu | 1 |
|            | Linear     | 256             | 256              | Relu       | 3 |
|            | Linear     | $256 + L_\mathbf{x} + m$ | 256 | Relu | 1 |
|            | Linear     | 256             | 256              | Relu       | 3 |
| $\sigma_\theta$ | Linear | 256            | 1                | Relu       | 1 |
| $c_\theta$ | Linear     | 256             | 256              | –          | 1 |
|            | Linear     | $256 + L_\mathbf{d} + n = 408$ | 128 | Relu | 1 |
|            | Linear     | 128             | 3                | –          | 1 |

Table 1: **Architecture of $g_\theta$.**

| Layer Type | Kernel Size | Stride | Activation | Normalization | Output Dimension |
|------------|-------------|--------|------------|---------------|------------------|
| Conv       | $4 \times 4$ | 2     | LRelu      | Spec, IN      | 128 |
| Conv       | $4 \times 4$ | 2     | LRelu      | Spec, IN      | 256 |
| Conv       | $4 \times 4$ | 2     | LRelu      | Spec, IN      | 512 |
| Conv       | $4 \times 4$ | 1     | LRelu      | Spec, IN      | 1 |

Table 2: **Architecture of $D_\phi$.**

**Volume Rendering:** In addition to pixel color $\mathbf{c}_r$, volume rendering can also be used to render alpha maps and depth maps, i.e. an alpha value $\alpha_r$ and a depth $d_r$:

$$\alpha_r = \sum_{i=1}^{N} T_r^i \alpha_r^i \qquad d_r = \sum_{i=1}^{N} T_r^i \alpha_r^i \tau_r^i \tag{2}$$

**Implementation Details:** Our PyTorch implementation is based on the code from https://github.com/yenchenlin/nerf-pytorch.git and https://github.com/LMescheder/GAN_stability.git. To compute FID and KID we adapt the code from https://github.com/abdulfatir/gan-metrics-pytorch.git.

## 1.2 Baselines

### 1.2.1 HoloGAN

We use the official implementation provided by the authors[1] and adhere to their training protocol: We use the Adam optimizer and train with an initial learning rate of $0.0001$ for experiments with an image resolution of $64^2$ pixels, and $0.00005$ for higher resolutions. We use a batch size of 32, train for 50 epochs, and linearly reduce the learning rate after the first 25 epochs. We further use the identity regularizer and style discriminator loss for an image resolution of $128^2$ pixels and higher. The only exception to this official training protocol we did is for the Chairs dataset. Here, we found that adding the identity regularizer improves performance as it is necessary to achieve multi-view consistent results, albeit the image resolution being $64^2$. For the datasets Faces and Cats, we use the pose ranges published by the authors as input. For the additional datasets which were not used by the HoloGAN authors (Cars, Birds, Chairs), we use the same pose ranges as for our method as input.

In the official implementation, two generator architectures are provided for an output resolution of $64^2$ and $128^2$, respectively. To train HoloGAN on image resolutions of $256^2$ and $512^2$, we extend the architecture used for an image resolution of $128^2$ by one or two transposed convolutional layers with adaptive instance normalization and leaky ReLU activation, respectively, and, similar to the previous layers, and apply the style discriminator loss to these intermediate outputs as well.

HoloGAN Generator Architecture

| Layer Type | Kernel Size | Stride | Activation | Normalization | Output Dimension |
|---|---|---|---|---|---|
| UpConv | $3 \times 3 \times 3$ | 2 | LRelu | AdaIN | $8 \times 8 \times 8 \times 256$ |
| UpConv | $3 \times 3 \times 3$ | 2 | LRelu | AdaIN | $16 \times 16 \times 16 \times 128$ |
| 3D Transformation | - | - | - | - | $16 \times 16 \times 16 \times 128$ |
| Conv | $3 \times 3 \times 3$ | 1 | LRelu | AdaIN | $16 \times 16 \times 16 \times 64$ |
| Conv | $3 \times 3 \times 3$ | 1 | LRelu | AdaIN | $16 \times 16 \times 16 \times 64$ |
| Concatenation | - | - | - | - | $16 \times 16 \times (16 \cdot 64)$ |
| Conv | $1 \times 1$ | 1 | LRelu | - | $16 \times 16 \times 512$ |
| UpConv | $4 \times 4$ | 2 | LRelu | AdaIN | $32 \times 32 \times 256$ |
| UpConv | $4 \times 4$ | 2 | LRelu | AdaIN | $64 \times 64 \times 64$ |
| UpConv | $4 \times 4$ | 2 | LRelu | AdaIN | $128 \times 128 \times 32$ |
| UpConv | $4 \times 4$ | 1 | Tanh | - | $128 \times 128 \times 3$ |

HoloGAN w/o 3D Conv Generator Architecture

| Layer Type | Kernel Size | Stride | Activation | Normalization | Output Dimension |
|---|---|---|---|---|---|
| UpConv | $3 \times 3 \times 3$ | 2 | LRelu | AdaIN | $8 \times 8 \times 8 \times 256$ |
| UpConv | $3 \times 3 \times 3$ | 2 | LRelu | AdaIN | $16 \times 16 \times 16 \times 128$ |
| 3D Transformation | - | - | - | - | $16 \times 16 \times 16 \times 128$ |
| Concatenation | - | - | - | - | $16 \times 16 \times (16 \cdot 128)$ |
| Conv | $1 \times 1$ | 1 | LRelu | - | $16 \times 16 \times 512$ |
| UpConv | $4 \times 4$ | 2 | LRelu | AdaIN | $32 \times 32 \times 256$ |
| UpConv | $4 \times 4$ | 2 | LRelu | AdaIN | $64 \times 64 \times 64$ |
| UpConv | $4 \times 4$ | 2 | LRelu | AdaIN | $128 \times 128 \times 32$ |
| UpConv | $4 \times 4$ | 1 | Tanh | - | $128 \times 128 \times 3$ |

Table 3: **HoloGAN Generator Architectures.**

### 1.2.2 HoloGAN w/o 3D Conv

We further use a modified version of HoloGAN in our experiments. As evidenced by our experiments HoloGAN fails in disentangling viewpoint from appearance at high resolution, varying different style aspects like facial expression or even completely ignoring the pose input. We identify the learnable projection as the underlying cause for this behavior and therefore reduce its number of learnable parameters in our modified version. More specifically, we keep the same generator architecture except for the two 3D convolutional layers in the projection unit which we remove entirely. These two layers are used in HoloGAN after the rigid-body transformation and before the 3D to 2D projection. Please see Table 3 for a detailed comparison of the original generator and our modified version, both for an image resolution of $128^2$ pixels.

### 1.3 PLATONICGAN

PLATONICGAN encodes an input image into a latent code which is then fed into a generator to create a 3D volume. The 3D volume is rendered onto the 2D image plane using differentiable volumetric rendering. An adversarial loss is defined on the 2D images rendered at randomly sampled camera viewpoints. Further, PLATONICGAN adds a reconstruction loss (sum of squared differences) on generated and input image. In addition to 2D images, PLATONICGAN requires instance masks for training.

We use the official implementation provided by the authors[2]. For experiments on the Cars and Birds datasets we use the hyperparameters for ShapeNet from the original implementation. Specifically, we train the network with Wasserstein loss and gradient penalty [2], a batch size of 16, and a learning rate of 0.000001 and 0.0025 for discriminator and generator, respectively. The reconstruction loss is weighted with factor 100 and the convolutional layers in discriminator and generator use 128 feature

| Name | | Type | # Images | Resolution | Azimuth | Elevation | Radius | Near, Far | Field of View |
|---|---|---|---|---|---|---|---|---|---|
| Cars | [1] | Synthetic | 10000 | $64^2$-$512^2$ | 0°-360° | 0°-85° | 10 | 7.5, 12.5 | 30° |
| Chairs | [11] | Synthetic | 152680 | $64^2$ | 0°-360° | 0°-90° | 10 | 7.5, 12.5 | 30° |
| Faces | [6] [5] | Real | 202599 30000 | $64^2$-$128^2$ $256^2$-$512^2$ | 0°-90° | 70°-85° | 9.5-10.5 | 7.5, 12.5 | 10° |
| Cats | [16] | Real | 9407 | $64^2$ | 0°-70° | 70°-85° | 10 | 7.5, 12.5 | 10° |
| Birds | [13] | Real | 8444 | $64^2$ | 0°-360° | 60°-95° | 9-11 | 7.5, 12.5 | 30° |

Table 4: **Datasets and Hyperparameter Choices.**

channels. For the Chairs dataset, we change the learning rates of discriminator and generator to $0.000005$ and $0.00125$ to stabilize training. For both the Faces and Cats datasets, we reduce the learning rate of the generator further to $0.00075$ as otherwise we observe that training becomes unstable almost immediately.

To better match the viewpoint distribution of our training images, we sample camera poses on the upper hemisphere, while the original implementation uses the full sphere. Note, that changing this is not trivial as, due to the reconstruction loss, PLATONICGAN learns the 3D object in the *camera coordinate system* of the input image and not in the world coordinate system. To ensure that we sample camera poses on the upper hemisphere in the *world coordinate system*, we have to account for this coordinate transformation. For the Chairs and Cars datasets the ground truth camera pose of the images is known. In this case, we simply transform the learned 3D object back to the world coordinate system before we sample the camera viewpoint. For the Faces, Cats and Birds datasets this cannot be done as the ground truth camera poses are unknown. However, for Faces and Cats the camera poses of the images are in a very small range. Hence, we sample the camera pose in the camera coordinates for these datasets. For Birds, we follow the authors' suggestion and only sample camera poses with random azimuth but fixed polar angle.

## 2 Datasets

Table 4 lists all used datasets and the corresponding choices of our hyperparameters. Note, that we choose slightly different angular ranges for Faces and Cats compared to [9]. As our method adheres more closely to the given camera poses we use smaller ranges that are more consistent with the viewpoints we observe in the datasets. We now provide additional information for rendering and preprocessing.

**Cars:** For rendering the ground truth data, we create scenes with a single centered car using the Carla Driving simulator [1]. Camera poses are sampled uniformly on the upper hemisphere using 0°-360° and 0°-85° for azimuth and polar angle, respectively. The camera radius is set to $10$ and the field of view of the camera is 30°.

**Chairs:** We follow the rendering protocol from [10] and composite the images on white background.

**Faces:** Similar to [9], we crop the images around the center and apply random horizontal flipping to the images during training. We crop the images to size $108^2$ and $650^2$ pixels for celebA and celebA-HQ, respectively.

**Cats:** We follow the preprocessing protocol from https://github.com/microe/angora-blue/blob/master/cascade_training/describe.py for cropping the cat faces.

**Birds:** We use the instance masks by Ryan Farrell[3] to composite the birds on white background. Similar to [4] we remove images with less than 7 visible keypoints. Further, we make use of the instance masks to filter the images. Firstly, as the provided masks are not binary but rather contain confidence values, we discard images in which more than $25\%$ of the mask have a confidence lower than $85\%$. Next, we discard images in which the mask occupies more than $1\%$ of the image boarder to avoid cropped close ups from birds and remove images in which the masks are split into multiple parts, to avoid occluded objects.

# 3 Results

## 3.1 Ablations

| Pos. Enc. | | FID | |
|:---:|:---:|:---:|:---:|
| $L_{\mathbf{x}}$ | $L_{\mathbf{d}}$ | Cars | CelebA |
| 60 | 24 | 41 | 35 |
| 30 | 12 | 54 | 44 |
| – | – | 77 | 109 |

Table 5: **Positional Encoding.** Ablation study comparing different dimensions used in the positional encoding in terms of FID score.

| Image Size | Time per Image $t$ [s] |
|:---:|:---:|
| $64^2$ | 0.1 |
| $128^2$ | 0.4 |
| $256^2$ | 1.6 |
| $512^2$ | 6.2 |

Table 6: **Runtime at Inference** for generating a single frame at different image resolutions.

**Positional Encoding:** The FID scores in Table 5 suggest that our approach benefits from the positional encoding $\gamma(\cdot)$. This finding is in agreement with the results from [8]. In particular for Faces, this embedding is crucial as this dataset contains many high-frequency details. We observe that otherwise the discriminator quickly becomes too strong leading to mode collapse, even when reducing its learning rate by a factor of ten.

**Runtime at Inference:** We measure the runtime for synthesizing a single image on a GeForce GTX 1080 Ti with batches of 8 images. The results in Table 6 show, that the runtime grows approximately linearly with the number of pixels, i.e., grows by a factor of 4 when both image width and height are doubled.

## 3.2 COLMAP

We evaluate the multi-view consistency of different methods by running a Multi-View-Stereo (MVS) algorithm on the generated images. The reconstruction quality reflects the multi-view consistency as the dense reconstruction relies on consistency across views. The MVS algorithm requires camera poses as input and we apply a Structure-from-Motion (SfM) algorithm to estimate the camera poses from the rendered images. This ensures a fair comparison to HoloGAN as the rendered images of HoloGAN do not necessarily stick to the input camera poses due to its learnable projection. We use COLMAP [12] for both, SfM and MVS estimation.

We conduct experiments on the Cars and Faces datasets at an image resolution of $256^2$ pixels. For Cars, we generate 400 images from the trained models for each instance using randomly sampled camera poses. We use 200 images for each face instance considering its smaller viewpoint range. Fig. 1 illustrates the camera poses estimated by the SfM algorithm using images from different methods on Cars. Our generated images allow for accurate pose estimation while the pose estimation for both HoloGAN variants fails on a large number of frames due to incorrect correspondences across different images. Similar to our observation on 2D images, HoloGAN w/o 3D leads to better pose estimates compared to the original HoloGAN. On face images, as shown in Fig. 2, the estimated camera poses are very far away from the object for both Hologan variants and do not recover the full azimuth range used for generating the input images. In contrast, the synthesized images from our method allow for recovering the correct camera distribution.

We further show the reconstructed point clouds for Cars and Faces in Fig. 3 and Fig. 4, respectively. As illustrated in Fig. 3a, the original HoloGAN leads to either an extremely sparse point cloud or a planar object on the Cars dataset. The latter is caused by providing multiple nearly identical images as input to COLMAP. This is evidenced in Fig. 6 of the main paper where the input poses can be completely ignored in HoloGAN for Cars at image resolution $256^2$. For HoloGAN w/o 3D on Cars in Fig. 3b, the MVS algorithm fails to reconstruct the full object due to the limited number of estimated poses. The partially reconstructed object is also noisy as can be seen from the rotated point cloud. In contrast, our generated images allow for complete and accurate 3D reconstruction of the cars as shown in Fig. 3c. We observe that COLMAP sometimes struggles to distinguish the left and the right side of the car and reconstructs a car missing one side (bottom row in Fig. 3c). However, this is due to the symmetric structure of the car instead of inconsistent multi-view images.

(a) HoloGAN        (b) HoloGAN w/o 3D        (c) Ours

Figure 1: **Camera Poses** estimated on synthesized Car images using COLMAP. The input camera poses for generating the corresponding images are sampled randomly within the angular range of the dataset. The pose estimates of our generated images correctly recover the distribution of the input camera poses.

(a) HoloGAN        (b) HoloGAN w/o 3D        (c) Ours

Figure 2: **Camera Poses** estimated on synthesized Face images using COLMAP. The input camera poses for generating the corresponding images are sampled randomly within the angular range of the dataset. The pose estimates from our generated images correctly recover the distribution of the input camera poses.

Fig. 4 suggests that COLMAP struggles to reconstruct a plausible face model from both HoloGAN variants. It instead estimates a planar structure or a noisy point cloud for each face instance as can be seen from the rotated point cloud. The result suggests that our model has a stronger inductive bias to learn a 3D structure. In contrast, HoloGAN generates plausible images from different viewpoints, but they do not correspond directly to the underling 3D structure. This also results in noise in the reconstructed point clouds in Fig. 4a and Fig. 4b. Our method instead leads to a smooth 3D point cloud with a plausible shape of a human face.

### 3.3 KID

In addition to the FID values in the main paper, we report KID values from the comparison to the baseline methods in Table 7 at resolution $64^2$ and in Table 8 at higher resolutions. For all experiments the KID values behave similarly to the corresponding FID values discussed in the main paper.

### 3.4 Viewpoint Controllability

We provide additional results on viewpoint controllability for our method at image resolution $64^2$ pixels in Fig. 5, Fig. 6 and Fig. 7. We show images and depth maps at resolution $256^2$ and $512^2$ pixels in Fig. 8, Fig. 9, Fig. 10 and Fig. 11. On all datasets our method is able to disentangle object identity and camera viewpoint and even learns reasonable depth maps.

(a) HoloGAN

(b) HoloGAN w/o 3D

(c) Ours

Figure 3: **3D Reconstruction** from synthesized images at resolution $256^2$. We show one example of the generated images on the left column. The remaining columns show the reconstructed point cloud from different viewpoints.

(a) HoloGAN

(b) HoloGAN w/o 3D

(c) Ours

Figure 4: **3D Reconstruction** from synthesized images at resolution $256^2$. We show one example of the generated images on the left column. The remaining columns show the reconstructed point cloud from different viewpoints.

| | Chairs | Birds | Cars | Cats | Faces |
|---|---|---|---|---|---|
| 2D GAN [7] | 3.15 | 0.68 | 3.76 | 0.27 | 0.28 |
| PLATONICGAN [3] | 18.8 | 18.6 | 13.6 | 41.8 | 44.2 |
| HoloGAN [9] | 3.02 | 5.64 | 9.70 | 0.98 | 1.17 |
| Ours | 0.96 | 2.61 | 0.91 | 0.84 | 1.17 |

Table 7: **KID**$\times 100$ at image resolution $64^2$.

| | Cars | | | Faces | | |
|---|---|---|---|---|---|---|
| | 128 | 256 | 512 | 128 | 256 | 512 |
| HoloGAN [9] | 16.47 | 17.80 | – | 2.38 | 5.20 | – |
| w/o 3D Conv | 13.33 | 14.95 | 23.32 | 1.65 | 1.59 | 2.98 |
| Ours | 1.78 | 4.03 | 5.52 | 2.03 | 3.66 | 3.74 |
| upsampled | – | 6.24 | 9.93 | – | 5.84 | 7.76 |
| sampled | – | 4.34 | 7.08 | – | 3.63 | 4.50 |

Table 8: **KID**$\times 100$ at image resolution $128^2$-$512^2$.

(a) Rotation      (b) Elevation

Figure 5: **Camera Pose Interpolations** for Chairs at image resolution $64^2$ pixels.

(a) Rotation      (b) Elevation

Figure 6: **Camera Pose Interpolations** for Birds at image resolution $64^2$ pixels.

(a) Rotation      (b) Elevation

Figure 7: **Camera Pose Interpolations** for Cats at image resolution $64^2$ pixels.

(a) Rotation

(b) Elevation

Figure 8: **Camera Pose Interpolations** for Faces at image resolution $256^2$ pixels.

(a) Rotation

(b) Elevation

Figure 9: **Camera Pose Interpolations** for Faces at image resolution $512^2$ pixels.

(a) Rotation

(b) Elevation

Figure 10: **Camera Pose Interpolations** for Cars at image resolution $256^2$ pixels.

(a) Rotation

(b) Elevation

Figure 11: **Camera Pose Interpolations** for Cars at image resolution $512^2$ pixels.

## 3.5 Random Samples

We provide randomly sampled images from our model for all datasets at resolution $64^2$ pixels in Fig. 12. The results indicate that our model is able to generate a large variety of instances for each dataset.

## 3.6 Limitations and Failure Cases

Lastly, we discuss failure cases and limitations of our approach. Fig. 13 shows two failure cases we observe in our generated images. Firstly, on the Birds dataset the model sometimes fails to learn the back of the bird correctly and instead adds a white patch on its back, see Fig. 13a, because the dataset contains only few images from the back view. This could be prevented by using instance masks as additional input channel to our discriminator. Secondly, on Cats the model learns an incorrect depth for the cat faces resulting in the face being oriented "inwards". The images in the Cats dataset only show little angular variation and, due to cropping the cat faces, are close to an orthographic camera projection. This makes depth estimation difficult and can create ambiguity, e.g., the rotation in Fig. 7 looks reasonable, albeit the incorrect depth. To resolve this issue, ground truth and generated depth maps could be added as input to the discriminator.

Our approach is currently limited to scenes with single objects. The failure cases above suggest that our approach could be extended to more complex scenes by adding more 2D supervision, e.g., depth maps from stereo depth estimation, or by leveraging symmetry constraints similar to [14]. We plan to analyze these effects in future work. Another interesting direction for future work is to reduce the time needed for rendering during inference.

## Footnotes

[1]https://github.com/thunguyenphuoc/HoloGAN

[2]https://github.com/henzler/platonicgan

[3]https://students.cs.byu.edu/~farrell/