[Reviews · NeurIPS 2020]

Review 1

Summary and Contributions: This paper builds on top of NeRF (Mildenhall et al. 2020) to create a generative model that learns to generate entire classes of objects from a training dataset, whereas NeRF focuses on fitting one single instance. The novel contributions for this paper include (1) creating a conditional version of NeRF that can be conditioned by a shape and appearance latent. (2) designing a clever 2-head architecture that disentangles appearance feature (from the color head + view direction vector) and the shape feature (from the volume density head) (3) shows that the model achieves better generated image quality and 3D consistency compared to baselines.

Strengths: There are various aspects that I like about the paper: (1) The design of the 2 head architecture to disentangle shape and appearance is clever. (2) The authors are very systematic with answering a range of research questions in their experimental evaluation section: comparing voxel representation to continuous representation, study the scalability of the approach to higher resolutions, discuss the pros and cons of learned neural rendering versus directly learning color values in 3D combined with differentiable ray-marching based rendering, studying the effects of multi-scale patch discriminator, and the quality of disentanglement between shape and appearance.

Weaknesses: (1) Though I acknowledge this work is of pretty good quality, the novelty is somewhat limited from a conceptual standpoint. This is a very straightforward extension of Mildenhall et al.: basically NeRF + GAN. It took a fairly incremental step for NeRF to extend it from a single object / instance to an entire class of shapes. (2) There is not sufficient experimental comparisons to Scene Representation Networks (SRNs) by Sitzmann et al., 2019. In my understanding, that paper also learns continuous shape representations rendered via differentiable ray-marching for entire class of objects (though they don't learn radience fields as a function of view direction). (3) This is a suggestion. The authors should comment on the possibility of abuse of this technology for applications such as DeepFakes that might present a challenge to society regarding fake new / misinformation etc. in the Broader Impact section.

Correctness: Yes the claims and methods are sound and credible.

Clarity: The paper is clear and well written. I find the schematics and figures very illustrative and informative.

Relation to Prior Work: Since the authors made detailed comparisons between voxel grid based 3D representations and continuous representations, I think the authors should discuss about hybrid continuous grid representations in the recent literature, e.g., - Jiang et al. Local Implicit Grid Representations for 3D Scenes. - Peng et al. Convolutional Occupancy Networks. - Chabra et al. Deep Local Shapes: Learning Local SDF Priors for Detailed 3D Reconstruction

Reproducibility: Yes

Additional Feedback:


Review 2

Summary and Contributions: The paper introduces Generative Radiance Fields (GRAF) for high-resolution 3D-aware image synthesis. Radiance fields have recently been proven successful for novel view synthesis of a single scene. Experimental results demonstrate the effectiveness of GRAF for high-resolution 3D-aware image synthesis.

Strengths: 1. While radiance fields show effectively representing 3D shapes and textures, the proposed GRAF can successfully capture high-resolution 3D-aware image synthesis from unposed images. 2. The introduced patch-based discriminator is effective. 3. The paper is well written.

Weaknesses: 1. It seems the proposed GRAF share similar concepts and frameworks with [41] for 3D-aware texture synthesis. However, the reference has not been adequately discussed in the paper. What are the key differences in the emphasized contributions between the two approaches? What are the differences between radiance fields and texture fields in [41]? Besides, the reviewer is interested in the qualitative and quantitative comparisons between the two methods. 2. Utilizing multi-scale patch discriminators may be an over-claimed contribution, since the strategy has been widely adopted in Image-to-Image translation tasks (e.g., CycleGAN and Pix2Pix) for high-resolution image synthesis. A minor concern: Does GRAF only have a GAN loss with an R1-regularization loss term?

Correctness: Yes.

Clarity: Yes,

Relation to Prior Work: Lack of comparisons and detail discussions with a closely related work [41].

Reproducibility: No

Additional Feedback: My major concern is the proposed "Radiance Fields" shares similar concepts and frameworks with "Texture Fields" [41] for 3D-aware texture synthesis. I have been convinced by the authors that the settings between these two are different. However, the reviewer still holds that using multi-scale patches is an over-claimed contribution. Overall, the proposed method is novel and effective for multi-view synthesis. I am willing to raise my rating to 6.


Review 3

Summary and Contributions: This paper presents a generative model for 2D images of objects that enables control of the object pose and camera viewpoint. It builds off of the HoloGAN [35] framework but replaces the 3D representation and forward rendering model with the more principled method from NeRF [33]. Comments after rebuttal/discussion: Although the novelty is a bit limited as it mostly uses the NeRF model in the HoloGAN framework, I believe that the results improve substantially upon HoloGAN and the idea is well executed and I stand by my original opinion that this paper should be accepted.

Strengths: The overall approach makes sense, and seems to fix many of the issues with the original HoloGAN work by using the more principled representation and rendering from NeRF. The presented results look quite impressive and suggest that the model from NeRF can be quite effective for generative models instead of just view interpolation.

Weaknesses: The novelty is slightly limited, as this paper mostly feels like a combination of HoloGAN and NeRF. However, I think that this is not a deal-breaker, and I appreciate the authors' efforts to distill the experiments into key takeaways that can be relevant for future progress in 3D-aware generative modeling. I am surprised about the ray sampling used in the patch discriminator. Since the same discriminator views different decimated versions of the image, I am surprised that the effects of aliasing do not prevent learning high-frequency details (I would be less concerned if different discriminators were used for different levels of subsampling, but in the proposed method, the same discriminator sees differently aliased versions of the same content). Maybe this would be relevant when trying to scale up to even higher resolutions? The presented results don't seem to showcase or take advantage of NeRF's ability to represent nice view-dependent reflection effects and specularities (and if I recall correctly, the Photoshapes dataset used in this paper should have shapes with realistic materials). I suggest that the authors comment on this and discuss what would be needed to scale this approach up to generate more realistic appearance. The generated faces in the supplementary video have interesting "ripple" effects/artifacts. I wonder what could be causing that?

Correctness: Yes, I believe that the paper's claims, algorithm, and experiments are correct and sound.

Clarity: Yes, the paper is well-written and generally easy to read and understand.

Relation to Prior Work: Yes, I believe that this paper adequately presents and discusses prior work.

Reproducibility: Yes

Additional Feedback: In 3.2.1, the text mentions that a uniform distribution on the upper hemisphere is used for camera location. Is this also true for datasets like faces and cars? If not I suggest clarifying.


Review 4

Summary and Contributions: This paper proposes a combination of gans and the neural radiance field (NERF). Using NERF as a internal representation, gans can generate view-consistent images. It inherits the advantages of implicit 3d representations but also can be used in a generative purpose, with disentangled view/color/shape codes. ##UPDATE: author response has addressed my concerns. I stand by my initial positive opinion.

Strengths: It is interesting to see NERF extended to consume unposed images and to a generative setting with disentangled shape/color codes. I believe this contribution is novel and timely, meeting current interest of the vision community.

Weaknesses: Since the view-consistency is claimed to be the main focus, it should be further evaluated in addtion to case demonstrations. One way to evaluate quantitatively is to report the FID of reconstructed shapes from generated multi-view images.

Correctness: Basically I think the proposed method is sound. But I'm concerned that the visual quality degrades compared to original NERF, as well as the view consistency. My view is that originally NERF overfits only one scene, so it does not need any constraint on the radiance field representation. However in the generative case it should have the ability to generalize, which may need additional inductive bias on the radiance field representation because a naive FC network is unlikely to generalize in a visually compelling way. Since this paper does not address this problem, the visual quality drops significantly. Another technical issue is that, the hand-crafted camera pose distribution seems to be incorrect for real datasets. This is likely to be one of the reasons that the novel view synthesis results of faces/birds shown in the supp. video are out of shapes. Did you try to learn pose distribution instead of hand-crafting for real datasets, say, sampling from Gaussian and use a small FC network to learn to map it into the pose space?

Clarity: yes.

Relation to Prior Work: yes.

Reproducibility: Yes

Additional Feedback:

[Author Response · NeurIPS 2020]

We thank the reviewers for their constructive comments. We appreciate that the reviewers find our contribution novel
and timely (R4), our architecture clever (R1) and successful (R2), and our results impressive (R3). R1 and R3 highlight
our systematic evaluation and all reviewers agree that the paper is clear and well-written and that the claims and method
are correct. In the following, we address the comments of the reviewers.

**Reviewer 1**

**Comparison to SRN:** SRN is a method for novel view synthesis of given objects while our model allows to generate
novel objects. The latter is not possible for SRN: SRN optimizes the latent code for a particular input image, but does
not provide a full probabilistic generative model for drawing unconditional random samples. It is non-trivial to extend
SRN to a full probabilistic generative model in order to conduct a comparison to GRAF. Moreover, SRN requires posed
images for training which are not available in our setting. We will clarify these differences in the final version.

**Broader Impact:** We thank the reviewer for the suggestion. We will discuss potential dangers of 3D-aware generative
models that can create 3D-consistent fake images. Such models might increase credibility of fake contents and
potentially fool systems that rely on multi-view consistency, e.g., modern face recognition systems.

**Hybrid representations:** We thank the reviewer for pointing out additional related work. We will extend our related
work section accordingly. While these works require 3D input and do not consider texture, extending our work to hybrid
representations is indeed an interesting avenue for future research.

**Reviewer 2**

**Comparison to Texture Fields:** Texture Fields map a *3D surface point* to a color value. Radiance fields take a 3D
point and a viewing direction as input and predict a color and a volume density for *any 3D point in space*. Importantly,
Texture Fields require a 3D shape as input (main paper, l.64) and colored surface points as supervision, while GRAF
learns from 2D supervision only. Furthermore, Texture Fields only allow for synthesizing novel textures conditioned on
a particular input 3D shape while we learn a generative model for both shape and texture. Given the different settings, a
fair comparison to Texture Fields is hence not possible. We will clarify these differences in the final version.

**Multi-scale patch discriminator:** We acknowledge that a patch discriminator with a *fixed* receptive field was previously
used for GANs (main paper, l.176). Both Pix2Pix and CycleGAN use a patch discriminator to model high-frequency
details but still require an additional loss on the full image. In contrast, we propose to use *multi-scale* patches to model
both local and global content, thereby avoiding to generate the full image which is difficult for neural radiance fields
due to the large computational and memory requirements involved in rendering a single pixel.

**Loss function:** We indeed train GRAF only with a non-saturating GAN loss and an R1-regularization (see Eq.(10)).

**Reviewer 3**

**High-frequency details:** Ideally, the generator learns to model high-frequency details to generate realistic patches
at the finest scale $s = 1$. We agree that it is indeed interesting to use scale dependent discriminators. To circumvent
excessive memory requirements we instead tried to append the scale to the discriminator input but did not observe an
improvement in our experiments. However, this might become useful when scaling to very high resolutions.

**Reflection effects and specularities:** We render Photoshapes with a big area light. Thus, the ground truth dataset does
not have strong reflections and specularities. For cars, our method is able to produce view-dependent specularities as
can be seen in the supplementary video (1:50-1:55). To further improve appearance it could be beneficial to disentangle
scene properties like materials and lighting in order to generalize to different lighting effects across scenes.

**Ripple artifacts:** We hypothesize that the ripple artifacts are a consequence of the positional encoding because it
transforms the input to a periodic signal. This naturally encourages periodic structures in the output.

**Reviewer 4**

**Quantitative evaluation of 3D consistency:** We thank the reviewer for
the valuable suggestion. We found that the Fréchet point cloud distance
[Shu et al., ICCV2019] is very sensitive to object poses. Therefore we adopt

| Method | Ours | HGAN | HGAN~~3D~~ |
|---|---|---|---|
| MMD-CD | **0.044** | 0.109 | 0.092 |

Minimum Matching Distance (MMD) [Achlioptas et al., ICML 2018] to measure the chamfer distance (CD) between
100 reconstructed shapes and their closest shapes in the ground truth. Results on cars suggest that our multi-view
consistent images lead to better 3D reconstruction compared to HoloGAN. This table will be added to the final version.

**Visual quality:** We thank the reviewer for suggesting to incorporate additional inductive biases. While our method
works well on simple objects like cars, we agree that more inductive biases are needed for complex real world scenes.
An interesting direction could be hybrid representations that combine convolutional architectures with FC networks.

**Learned pose distribution:** We thank the reviewer for the suggestion to also learn the camera pose distribution. We
indeed consider this as an interesting research direction and will explore it in our future work.

[Meta-Review · NeurIPS 2020]

In general, the reviewers were positive about the paper: the proposed GRAF can successfully capture high-res 3D-aware image synthesis from unposed images, the patch-based discriminator is effective and the paper is well written, while there was concern that the paper felt like a combination of HoloGAN and NeRF. After the rebuttal and discussion, the reviewers all voted for acceptance. Please put the important points in the rebuttal to the final version.